# OpenReview forum: "Evaluating Language Model Agency Through Negotiations"
_ICLR.cc/2024/Conference — ICLR 2024 poster_

### Official Review · Reviewer_ZZpj · 2023-10-24

**Soundness:** 2 fair
**Presentation:** 2 fair
**Contribution:** 2 fair
**Rating:** 5
**Confidence:** 4

**Summary:**

This paper considers using negotiation games to evaluate the intelligence of LLMs. The authors designed a specific structured negotiation protocols, where the agents need to compose a private mental note as well as a public message to the other party each negotiation round. The authors found that GPT-4 is generally more skillful in these negotiation games.

**Strengths:**

The idea of using negotiation game to assess the intelligence behaviors are promising.

**Weaknesses:**

The presentation is not clear. The evaluation studies are limited.

**Questions:**

I think the idea of using negotiation games to evaluate LLM is great. However, I have concerns about the approaches and evaluations in this paper. Also some of the concepts are not explained well. Specifically:

1. About the negotiation protocol & the way it uses LLM to compose a strategy. Where do $q_{n/m}, \beta_i$ and context $c$ come from? Are they fixed or sampled from some distribution during each negotiation instance? Why the negotiation strategy has to be constructed in such way, and how does it compare with other approaches? E.g., just directly input previous negotiation rounds results and output a text message.

2. Can the author further clarify what is distributive v.s. compatible negotiation?

3. For the cross-play results, are the scores in Table 5 and 6 averaged across every possible opponents? From Table 13, 14 it appears there certain strategic structure (such rock-paper-scissors cycle). What will be the mean conclusion from there then?

4. In you opinions, why different LLMs behaviors qualitively different?

5. There have been several previous works that evaluate LLMs using negotiation games [1, 2]. Can the authors compare your work with theirs.

[1] "Improving Language Model Negotiation with Self-Play and In-Context Learning from AI Feedback" Fu et. al.
[2] "Evaluating LLMs with Interactive Multi-Agent Negotiation Games", Abdelnabi et. al.

---

> ### Author Response · Authors · 2023-11-15
> **Response to Question 1**
>
> Thank you for taking the time to review our work and the thoughtful questions and suggestions. We hope that our answers below can clarify some of the perceived shortcomings in our presentation. We would like to point out that the presented evaluations comprise thousands of multi-turn negotiations, covering different game settings and models from all leading LM providers. Due to space constraints and to not overwhelm the reader, we did our best to compress our findings as concisely as possible. We will add additional analyses to the Appendix.
>
> > 1. About the negotiation protocol & the way it uses LLM to compose a strategy. Where do q_n/m and beta_i  and context c come from? Are they fixed or sampled from some distribution during each negotiation instance? Why the negotiation strategy has to be constructed in such way, and how does it compare with other approaches? E.g., just directly input previous negotiation rounds results and output a text message.
>
> The context, c, consists of the protocol, game, and issue descriptions. The protocol descriptions, consisting of the message and note query prompts, q_m and q_n, the negotiation rules, and termination conditions are fixed across negotiations. The rules used in our experiments are strongly inspired by those of the famous ‘Rio Copa’ negotiation game [1], a negotiating protocol commonly used to simulate negotiations for didactic purposes at leading business schools.
>
> The queries q_m and q_n are designed to elicit structured outputs and promote measurements of metrics relevant to alignment while remaining as concise as possible. These prompts are the results of many repeated experiments on held-out data across models to minimize promoting unintended behaviors. As noted in Section 5, the need for ‘prompt engineering’ is a limitation inherent to current language models. It is thus likely that a set of prompts exists that is even more effective than the ones used in our work. However, as all models use the same prompts, the only point of concern would be if the prompts would provide an unfair example to one of the models. Between the minimality of the prompts used and the extensive tests conducted, we did not observe any evidence that this is the case. In Appendix E.1 we listed the fixed rules and prompts used for our experiments.
>
> A game and issue descriptions and corresponding preference weights beta_i are fixed as well, based on the evaluator’s objectives. This basic formulation allows for a wide range of possible negotiations. The game and issues used in our experiments are described in Appendix E.2. For the non-integrative setting, issue preference weights beta_i are all equal, i.e. 1 / #-issues. For the integrative setting, the main requirement for the choice of preference weights is that clear collaborative bargaining opportunities become available, i.e., at least one issue is more important than the other issues. In our experiments, we therefore opted for a fixed {1/3,  2/3} v. {2/3, 1/3} split to provide clear trade-offs between issues. We will update the Appendix to include these experimental setup details.
>
> Generally, there are two ways a negotiation can be formatted using LMs:
> 1. **Direct dialogue between two LMs**: Each LM-instance takes the ‘Human role’ to query the other LM-instance, responding in the ‘AI role’. However, a third, ‘System’ role is necessary to initialize the negotiation rules and provide per-round instructions.
> 2. **Represent negotiations as a transcript**: Each LM-instance receives messages from the other LM-instance as a transcript presented by a ‘Human’ or ‘System’ role. This same Human/System role can be used to initialize and mediate the negotiations, thus only requiring two distinct API roles.
>
> As listed in Appendix B, not all LM-providers currently support three distinct roles (AI, Human, and System). Following preliminary experiments described in Appendix A.2 to measure the difference between the two formats, we opted to use the ‘transcript’ format to include models from all popular LM-providers.
>
> As described in equations 2, 3, and 4, when formulating its next note/message, the LM-agent indeed observes a sequence of the previous negotiation rounds (note the t superscripts). As agents are initialized using the context c, these were omitted in these equations. However, we recognize that this notation is not clear enough and will update the revised text accordingly. Thank you for pointing this out and we apologize for the confusion.
>
> [1] Bontempo, R., & Iyengar, S. (2008). Rio Copa: A negotiation simulation. Columbia Caseworks.

---

> > ### Author Response · Authors · 2023-11-15
> > **Responses to Questions 2-5**
> >
> > > 2. Can the author further clarify what is distributive v.s. compatible negotiation?
> >
> > Let us try to explain the different issues using the following analogy: two friends are ordering a pizza to share. They have to decide on how to divide the pizza slices, and on the amount of cheese to put on the pizza.
> > - Distributive negotiation: The number of slices represents a fixed resource, where distributing one slice to friend A automatically means one slice less for friend B. This is a distributive or ‘zero-sum’ issue.
> > - Compatible negotiation: On the other hand, both friends might equally love cheese. Hence, the issue is compatible with both friends’ objectives.
> >
> > We discuss the different types of issues and games in Section 2.1, “Cooperative v. Competitive Performance”, but will update the revised text to make these descriptions clearer.
> >
> >
> > > 3. For the cross-play results, are the scores in Table 5 and 6 averaged across every possible opponents? From Table 13, 14 it appears there certain strategic structure (such rock-paper-scissors cycle). What will be the mean conclusion from there then?
> >
> > For the cross-play evaluation, each model plays against the other models for each game setting multiple times with a temperature of 0.2. The results per game setting are then averaged across all opponents.
> >
> > The performance of some models might indeed be very close to each other. Reporting win-shares over-discretizes the metrics of interest which could lead to the appearance of ‘rock-paper-scissors’ like structures. Generally, models that perform significantly better, e.g., gpt-3.5-turbo, perform better against most models. Increasing the number of negotiation runs and scenarios will further improve predictive performance. As mentioned in Section 5, we imagine the use of latent-ability frameworks like the ELO rating system to extrapolate ranking results across models. In such a system, it might be possible that weaker models outperform stronger models occasionally, but not consistently.
> >
> > Since our submission, we have complemented our results with additional experiments and analyses. For example, we moved away from reporting ‘win-shares’, and instead opted for displaying normalized utility and normalized utility for completed games, U and U* respectively. This makes it easier to compare self-play and cross-play results and reduces the unnecessary inflammatory connotation of the term ‘win-shares.’ Tables 5 and 6 in the main text and Tables 13 and 14 in the Appendix will be updated accordingly in the revised text.
> >
> > > 4. In you opinions, why different LLMs behaviors qualitively different?
> >
> > That is a great question. Understanding the qualitative differences in LM behavior is currently an open question in the field, not isolated to our work on negotiations. The combination of different architectural choices, massive amounts of data used during pre-training, and various finetuning methods (e.g., RLHF), makes causal statements about behavior challenging. Unfortunately, opaque model training details of commercial models exacerbate the complexity of this problem even further. A useful first step would be increased transparency about the training data used, combined with research efforts to find direct links between training data and inference behavior.
> >
> > > 5. There have been several previous works that evaluate LLMs using negotiation games [1, 2]. Can the authors compare your work with theirs.
> > [1] "Improving Language Model Negotiation with Self-Play and In-Context Learning from AI Feedback" Fu et. al. [2] "Evaluating LLMs with Interactive Multi-Agent Negotiation Games", Abdelnabi et. al.
> >
> > To the best of our knowledge, the use of negotiation games as a construct to jointly evaluate performance and alignment does not yet exist in the literature. The first paper mentioned [1] is discussed at the end of our related work (Section 6), where we note that the authors focused on single-shot negotiation game settings. In their setup, single-shot, single-issue negotiation games were complemented by a memory of ‘past’ negotiations, hence resulting in limited single-session interactions and minimal game complexity. The focus of the authors was to measure if LM-agents could ‘train’ other LM-agents to improve their performance, whereas our focus is on using negotiation games to evaluate LM-agency both from an alignment and performance perspective. The second work [2] seems to have been submitted to the same conference as our paper (ICLR’24) and was thus not on our radar while developing this work. However, upon reviewing [2], their setup significantly differs from ours. For example, the authors focus on multi-agent round-table negotiations, not the joint evaluation of performance and alignment. It is further less clear how their setup enables seamless cross-model evaluations.

---

### Official Review · Reviewer_pM8H · 2023-10-30

**Soundness:** 2 fair
**Presentation:** 3 good
**Contribution:** 2 fair
**Rating:** 3
**Confidence:** 2

**Summary:**

- Joint framework to evaluate performance and alignment of LLMs using structured negotiation tasks.

- Creates a negotiation task benchmark, which involves evaluating the success of LLMs negotiating toward goals in self-play and cross-play.

- Incorporates LMs into the evaluation benchmark so that the benchmark "co-evolves" with the models they are designed to test.

Overall, I am currently giving this paper a 3 (reject) before discussions, considering the weaknesses outlined below related to empirical design and the lack of formalism. However, I am giving my rating a confidence of 2 since I am unfamiliar with related work.

**Strengths:**

- timely and relevant subject

- interesting idea on evaluating both alignment and performance, notably given the uncertainty around the orthogonality hypothesis.

- interesting cross-play results to compare LLMs to each other.

**Weaknesses:**

# Big weaknesses

- empirical design with insurmountable reproducibility issues and cost issues impacting statistical validity. Unless the experiments were performed all simultaneously, it is not obvious that they are valid since these models undergo continuous improvement, meaning you might've been comparing different models across different experiments, even if API access was the same and there would be no way to know, right? For this same reason, the experiments are not necessarily reproducible. It might be better in the future to use open-source LLMs for which the models can be held frozen by selecting a checkpoint.

- no definition of agency despite it being a central concept to the paper

- the metrics in table 1 seem to require a lot of human checking, which makes it difficult to scale this benchmark. Are you also using LLMs to compute these benchmark values(e.g. internal faithfulness)?

- there is a whole range of issues between opposing and aligned interests, e.g. mixed cooperative-competitive settings or variable-sum games. It would be interesting to establish benchmarks on these types of settings as well.

- having each agent play both sides and starting positions and averaging does not control for bias, since different LLMs might be more or less able to take advantage of these asymmetries depending on the game design. You would probably also need to ensure sufficient diversity in the game contexts to help control for bias (i.e. not just rent negotiation games)

- it is not obvious that allowing multiple turns to take place would provide more information into understanding which persona is active especially if the persona mixture depends on previous context and can evolve through a conversation, nor that this persona activation would be consistent across different runs with different random seeds. However, I am not very familiar with this literature.

===

# Small weaknesses

- bad reference formatting: "Jacob Andreas. Language models as agent models, 2022."

- fix typos (" the challenge is to figure out agent interests are aligned. command, ...")

- no reference provided for "Theory of Mind"

- a cooperative game in game theory is one in which players can negotiate binding contracts, which can be confusing given that we are discussing games in game theory formalism, though with a different meaning for "cooperative".

- not obvious that issues necessarily have linearly weighted preferences. A related subject is scalarization in multi-objective optimization.

- "the possible effects and opportunities of stories, traits, rules, and prompts have been discussed in the previous subsections" stories were not discussed

- "providing too much capacity might lead to hallucinations" citation needed

**Questions:**

- ToM strategy is not introduced formally

- Why is the utility 0 if there is no agreement on all issues?

- How are the prompts designed to test the LLM's negotiation capacities? For any given game, do you test multiple prompt variations with similar semantic meanings? How do you know the elicited personas will be the same across differences in the input prompts?

- Why is the goal to measure if there is a significant difference in performance between the average, expert and novice initialization? I thought the goal was to evaluate LLMs in general, and it's far from obvious that such an initialization will transfer the same way across different LLMs

- Does the co-evolution of the benchmark in terms of cross-play rely on having sufficient diversity among language models? How do you see the benchmark holding up in the future?

**Details Of Ethics Concerns:**

The author discusses the ethical considerations of their work in relation to malicious actors. AI governance is outside my area of expertise, therefore I am flagging for Ethics Review such that an ethics reviewer may review the author's claims in Section 5, "Ethical Considerations" paragraph, though I do not believe that this framework constitutes a potentially harmful methodology on its own, it is aimed at evaluating AI capabilities and alignment.

---

> ### Author Response · Authors · 2023-11-15
> **Response to Weaknesses 1-3**
>
> Thank you for your detailed review and for voicing your concerns. We hope that our answers to the perceived weaknesses you highlighted and the questions you posed will allow you to reassess our work. In addition to the replies below, we also included a general summary placing our efforts more squarely in the currently accepted approach to language model evaluation benchmarking.
> > 1. empirical design with insurmountable reproducibility issues and cost issues impacting statistical validity. Unless the experiments were performed all simultaneously, it is not obvious that they are valid since these models undergo continuous improvement, meaning you might've been comparing different models across different experiments, even if API access was the same and there would be no way to know, right? For this same reason, the experiments are not necessarily reproducible. It might be better in the future to use open-source LLMs for which the models can be held frozen by selecting a checkpoint.
>
> > a. [Experimental results might not be valid and are not necessarily reproducible since closed-source models are continuously improving]
>
> Language modeling (LM) technology has matured to a point where commercial applications are viable, leading to closed-source implementations to protect large R&D investments. The reviewer’s argument appears to be against evaluating closed-source models in general. We would argue that this strategy is dangerous. Especially now that LMs are rapidly entering the public sphere and are being used by hundreds of millions, more effort than ever should be dedicated to evaluating performance and alignment.
>
> Any evaluation benchmark is limited by how often evaluations are performed. For example, the current industry-leading LM evaluation benchmark, ‘HELM’ [1], has the same limitation. Furthermore, closed-source models often come with external checkpoints. More importantly, they surely come with internal checkpoints. Legislative pressure could force LM providers to share evaluation metrics on several benchmarks before public releases, similar to how other models deployed in critical infrastructure are treated, e.g., risk models for consumer lending at retail banks.
>
> [1] Liang, Percy, et al. "Holistic evaluation of language models." Transactions on Machine Learning Research (2023).
>
> > b. Cost issues [impact] statistical validity’
>
> Machine learning models have dramatically increased in size over the past 15 years with the rise of deep learning. This increase has come at a price of reproducibility, e.g., not many labs have access to hundreds of GPUs to host a single model. The recent growth of APIs for deep learning models has increased the possibility for evaluations by lowering the thresholds of accessibility. Yet, we recognize that performing evaluations on a large suite of models comes at a hefty price (The experimental costs for this work were ~USD 10,000-). As noted in the previous point, the cost of running such evaluations between models from large providers could be imposed on the large model providers themselves before public releases. Indeed, one of our core contributions to the community is the dataset of thousands of negotiation transcripts to serve as a starting point for further research.
>
> However, as outlined in Section 5, ‘Limitations and Ethical Considerations’, a viable alternative exists for smaller labs or individual researchers interested in benchmarking their models against third-party options. By establishing a latent-ability framework like the ELO rating system used in chess, a model could be compared to cheaper third-party models, after which results can be extrapolated.
>
>
> > c.  It might be better in the future to use open-source LLMs for which the models can be held frozen by selecting a checkpoint
>
> As described in Section 3.2 ‘Qualifiers’ and at the start of Section 4, we indeed attempted to benchmark the LLaMA 2 model family, widely considered the state-of-the-art in open-source LLMs at the time of research. However, we found that these models were unable to pass our Qualifier round, i.e., to successfully and consistently complete a structured negotiation.
>
> > 2. no definition of agency despite it being a central concept to the paper
>
> In the first paragraph of our introduction, we describe LM-agents as: ‘capable of completing tasks that require interactive reasoning [... and] acting over multi-step horizons.’ We will emphasize and sharpen this definition in the revised text.
>
> > 3. the metrics in table 1 seem to require a lot of human checking, which makes it difficult to scale this benchmark. Are you also using LLMs to compute these benchmark values(e.g. internal faithfulness)?
>
> The metrics displayed in all tables are fully automated, as described in Appendix A.4. We utilize a mixed strategy of regexes and an LM equipped with a concise set of hand-crafted examples. We further performed a large number of sampled spot-checks to ensure the extracted offers were reasonable and accurate.

---

> > ### Author Response · Authors · 2023-11-15
> > **Response to Weaknesses 4-6 and Small Weaknesses**
> >
> > > 4. there is a whole range of issues between opposing and aligned interests, e.g. mixed cooperative-competitive settings or variable-sum games. It would be interesting to establish benchmarks on these types of settings as well.
> >
> > You are quite right! An important motivation for using structured negotiations as a construct for dynamic evaluations is that they are easy and flexible to extend. Using the minimal atomic building blocks of ‘distributive’ and ‘compatible’ issues and preference weights to enable collaborative bargaining agreements already allows for a wide range of optimization problems. Indeed, in our two-issue game experiments, we report results for the mixed ‘cooperative-competitive’ setting. In Table 3, see columns (Non-) Integrative Mix., and in Table 5, these fall under ‘Two-issue games’ / ‘Cooperative’. We are excited to see the type of games the wider research community can come up with and welcome any suggestions you might have.
> >
> > > 5. having each agent play both sides and starting positions and averaging does not control for bias, since different LLMs might be more or less able to take advantage of these asymmetries depending on the game design. You would probably also need to ensure sufficient diversity in the game contexts to help control for bias (i.e. not just rent negotiation games)
> >
> > For a given game, averaging performance over sides and starting positions guarantees a debiased output. You are correct that this type of debiasing operation only solves intra-game bias. Similar to the real world, games and issues with different descriptions can be perceived as harder/easier by different agents - even if the underlying payoff matrices are the same. This could indeed lead to “inter-game” bias.
> >
> > Yet, as described in Section 2.1 “Modulating Complexity”, we see this as a feature of structured negotiations, not a bug. During the development of this project, we indeed experimented with various game settings, e.g., corporate mergers, loan agreements, etc. We are planning on including additional game settings and evaluation results in the Appendix. We would welcome suggestions you might have on game contexts to consider.
> >
> > > 6. it is not obvious that allowing multiple turns to take place would provide more information into understanding which persona is active especially if the persona mixture depends on previous context and can evolve through a conversation, nor that this persona activation would be consistent across different runs with different random seeds. However, I am not very familiar with this literature.
> >
> > Our goal is to evaluate LMs' ability to behave as agents, capable of consistently and reliably completing tasks over a series of steps according to some constraints. Our framework is designed to jointly evaluate this ability from an alignment and performance point of view. Thus, even if the persona-mixture changes during an extended period of interaction, sequence-level metrics provide more insight into how the average agent produced by such an LM would behave as opposed to a single observation. We further stabilize these metrics by averaging multiple runs with different random seeds. While analyzing persona-mixtures is not the central topic of our work, the various open questions in this area present interesting opportunities for future work.
> >
> >
> >
> > >## Small weaknesses
> >
> > > 1. bad reference formatting: "Jacob Andreas. Language models as agent models, 2022."
> >
> > Thank you for pointing out this reference.
> >
> > > 2. fix typos (" the challenge is to figure out agent interests are aligned. command, ...")
> >
> > Thank you for pointing out this typo.
> >
> > > 3. no reference provided for "Theory of Mind"
> >
> > We added a reference to the original work by Premack and Woodruff (1978) responsible for coining the term.
> >
> > > 4.  a cooperative game in game theory is one in which players can negotiate binding contracts, which can be confusing given that we are discussing games in game theory formalism, though with a different meaning for "cooperative".
> >
> > Thank you for this suggestion. We will take it into consideration.
> >
> > > 5. not obvious that issues necessarily have linearly weighted preferences. A related subject is scalarization in multi-objective optimization.
> >
> > We fully agree. Linearly weighted preferences are a choice that can be deviated from depending on the evaluator’s objectives.
> >
> > > 6. "the possible effects and opportunities of stories, traits, rules, and prompts have been discussed in the previous subsections" stories were not discussed
> >
> > Thank you for pointing this out and our apologies for the confusion: “stories” refers to the game and issue descriptions. We will update this in our revised text.
> >
> > > 7. "providing too much capacity might lead to hallucinations" citation needed
> >
> > Thank you for pointing this out. We will add a citation to the revised text.

---

> > > ### Author Response · Authors · 2023-11-15
> > > **Response to Questions**
> > >
> > > ## Questions:
> > > > 1. ToM strategy is not introduced formally
> > >
> > > Theory of Mind (ToM) is not introduced as a strategy (see Section 2.2). Instead, it is introduced in the context of a metric relevant to measuring state-of-mind consistency we call “external faithfulness”. External faithfulness is defined as the alignment between publicly made offers and the estimate of agreeable offers from the other agent’s perspective. For example, assume two friends are negotiating about how to best divide ten slices of a pizza. Friend A makes an offer of five slices to friend B. Then, providing the same context available to friend A before making the offer, we inquire what friend A believes friend B would consider an agreeable offer. This would be considered ‘Theory of Mind’ inference. If friend A believes friend B would have agreed to less than five slices of pizza, the publicly stated offer is not considered faithful.
> > >
> > > > 2. Why is the utility 0 if there is no agreement on all issues?
> > >
> > > This is a common choice in negotiation games that can easily be changed based on the evaluator’s objectives.
> > >
> > > > 3. How are the prompts designed to test the LLM's negotiation capacities? For any given game, do you test multiple prompt variations with similar semantic meanings? How do you know the elicited personas will be the same across differences in the input prompts?
> > >
> > > The prompts we used are designed to elicit structured outputs and promote measurements of metrics relevant to alignment while remaining as parsimonious as possible. These prompts are the results of many repeated experiments to minimize promoting unintended behaviors. As noted in Section 5, the need for ‘prompt engineering’ is a limitation inherent to current language models. It is thus likely that a set of prompts exists that is better suited for our stated goals than the ones we found. In Appendix E.1 we listed the fixed rules and prompts used for our experiments.
> > >
> > > > 4. Why is the goal to measure if there is a significant difference in performance between the average, expert and novice initialization? I thought the goal was to evaluate LLMs in general, and it's far from obvious that such an initialization will transfer the same way across different LLMs
> > >
> > > The primary goal of our work is indeed to present a framework to evaluate LM agency in general. As noted at the end of the introduction, we do not perform a comprehensive investigation of model-level parameters to optimize negotiating agents. We address the possibility of using different agent skill level initialization in our discussion of possible ‘Persona Bias’ (Sections 2.4 and 3.), the results of this limited analysis are presented in Appendix C.1, Tables 11 and 12. This analysis is performed in response to a commonly reported strategy of practitioners to improve agent performance by providing elaborate agent descriptions. The idea is that such initializations would conjure agents with the desired abilities. We therefore decided to include a limited analysis as a proxy to test the effect of skill initialization.
> > >
> > > > 5. Does the co-evolution of the benchmark in terms of cross-play rely on having sufficient diversity among language models? How do you see the benchmark holding up in the future?
> > >
> > > The cross-play benchmark is meant to evaluate LM-agent behavior in relation to LM-agents from other LM providers. We designed this benchmark to proactively prepare for a possible future where LM-agents from different providers are deployed in the real world. This benchmark can be used both to measure progress between models over time and to provide an up-to-date central evaluation source. The co-evolution of such an evaluation source would not necessarily rely on having diversity among LMs, but rather on its reflection of models deployed in the real world.

---

### Official Review · Reviewer_yVDW · 2023-10-31

**Soundness:** 3 good
**Presentation:** 4 excellent
**Contribution:** 3 good
**Rating:** 8
**Confidence:** 4

**Summary:**

The authors are proposing a technique to evaluate large language models using a scenario where they are required to participate in a multi-issue negotiation, for instance a rental agreement. The overall claim of the paper is that investigating such a negotiation might lead to a more accurate evaluation of the performance and alignment of the language model compared to other approaches. The authors had tested a number of currently available language models through their APIs.

**Strengths:**

* The authors are making a good case that the proposed evaluation method is a useful aspect of the behaviors of the large language models.
* The paper proposes a methodology that carefully considers the variety of biases that can be introduced by the measuring process, and takes credible steps to avoid them.
* Extensive evaluation over six-seven LLMs, including self-play and cross-play.

**Weaknesses:**

* Many of the current language models are not trained to sustain a negotiation type conversation. For instance, they don't have a framework to keep track of the issues agreement had been reached upon, or the current alternatives that are under discussion. Thus, the proposed metric measures an aspect on which the models had not been trained, and indeed their performance on it is more a side effect of some artifacts in the training data.

**Questions:**

Clearly, the performance of the LLMs in this task can be improved relatively easily, as the underlying mathematical negotiation problem is much simpler than LLM's language abilities. How would one rank a model that would have minimal language abilities, but use a specialized algorithmic plugin?

---

> ### Author Response · Authors · 2023-11-15
>
> Thank you for your positive and constructive review. We are happy to see our effort to de-bias the presented metrics and the extensive evaluations conducted being recognized. We hope the responses below can alleviate the remaining perceived weakness and answer your question.
>
> > Many of the current language models are not trained to sustain a negotiation type conversation. For instance, they don't have a framework to keep track of the issues agreement had been reached upon, or the current alternatives that are under discussion. Thus, the proposed metric measures an aspect on which the models had not been trained, and indeed their performance on it is more a side effect of some artifacts in the training data.
>
> Thank you for highlighting this important point. Current language models (LMs) are indeed not explicitly optimized for multi-step negotiations. Yet, LMs are broadly marketed as “AGI”, which encourages users to use them as agents for tasks not seen during training (see for example the recent “personal GPTs” release by OpenAI [1]). We’re testing to what extent such (encouraged) “off-label use” is warranted.
>
> As many of the challenges integral to successful negotiations show up in various real-world tasks of interest, negotiation performance can serve as an insightful proxy for expected behavior. Additionally, metrics like ‘internal/external faithfulness’ and instruction-following provide insight into the consistency of an LM-agent’s state of mind and the ability to stay within user-defined boundaries.
>
> [1] https://openai.com/blog/introducing-gpts
>
> > Clearly, the performance of the LLMs in this task can be improved relatively easily, as the underlying mathematical negotiation problem is much simpler than LLM's language abilities. How would one rank a model that would have minimal language abilities, but use a specialized algorithmic plugin?
>
> That is a great question. It is indeed possible that a new generation of specialized negotiation-algorithms will emerge, where LMs will be used as ‘sensing’ and ‘acting’ modules. Such a module would be responsible for decoding offer indications in natural language to a more restricted format for a specialized algorithmic solver, then encode back proposed solutions into natural language to communicate externally. We highlight examples of these types of ‘hybrid-LM-agents’ in our related work. Yet, even for such hybrid-models, the bar for ‘minimal language abilities’ may be quite high to accurately capture all subtleties contained in natural language.
>
> While it is clear a specialized solver would be much more capable of finding ‘optimal’ solutions to the underlying mathematical problem, it is less clear how much this would help in finding the language required to successfully convince the opposing agent toward an acceptable agreement. Additionally, strong language understanding would be required to ‘guess’ the other agent’s payoff matrix, i.e., perform “Theory of Mind” inference.
>
> However, the purpose of our work is not necessarily to provide a benchmark for the best ‘negotiating’ agent, but rather measure the innate, un-optimized ability of LMs to behave as agents. If it becomes the case that LM providers would find negotiating performance important enough to design specialized training/finetuning and inference routines, such advantages should quickly disappear in cross-play performance.

---

> > ### Comment · Reviewer_yVDW · 2023-11-23
> >
> > Thank you for the additional explanations. I will retain my rankings.

---

### Public Comment · ~Guohao_Li1 · 2023-11-14
**Suggesting related work**

The paper, "Evaluating Language Model Agency through Negotiations," proposes a novel framework to assess language model (LM) performance and alignment using structured negotiation games. This method addresses the dynamic and interactive nature of potential real-world LM applications, which current static evaluation methods fail to capture. The authors argue that negotiation games offer scalable, difficult-to-hack performance metrics and insights into model decision-making. The study involves extensive empirical testing of publicly available models from major providers like Anthropic, Cohere, Google, Meta, and OpenAI. The key findings include the inability of open-source models to complete negotiation tasks, the challenges posed by cooperative bargaining games, and the observation that the most powerful models do not always 'win'. The paper contributes by offering a new evaluation paradigm for evolving LM agency and releasing an open-source library to facilitate research in this area.

Thanks for the great work! It could also be beneficial to discuss prior work on multi-LLM agents for the study of cooperative AI [1].

[1] Li, Guohao, Hasan Abed Al Kader Hammoud, Hani Itani, Dmitrii Khizbullin, and Bernard Ghanem. "CAMEL: Communicative Agents for" Mind" Exploration of Large Language Model Society." NeurIPS 2023

---

### Author Response · Authors · 2023-11-15
**General Summary**

Dear Reviewers,

We would like to thank you for your time and feedback in reviewing our work. As a complement to the individualized rebuttals to your reviews, we offer a summary of our work's core purpose and contributions.

The use of language models (LMs) as agents presents a significant shift in paradigm away from stateless, question-answering applications. Current evaluation methods are not suited to evaluate LMs as agents due to being a product of the old paradigm. Specifically, they fail to measure LMs’ ability to autonomously plan and make decisions over several steps of interaction. Nor do they account for various alignment factors important for safe real-world deployment such as steerability, transparency, and robustness. In our work, we advocate for a shift in the community’s approach to evaluating LMs commensurate to the shift in paradigm:
1. Dynamic applications require dynamic evaluations to capture behavior relevant for deployment in the real world.
2. Alignment and performance metrics should be measured jointly to promote holistic, safe model development.
3. Evaluations should support the ability to measure cross-model interactions of different LMs.

In addition to these new evaluation requirements, we point out two known weaknesses of static evaluations:
1. Static evaluation tasks are notoriously vulnerable to ‘data leakage’ events, where models gain access to test data during training. Ideally, we would not want our tasks fixed in advance to avoid the possibility of data leakage.
2. The pace of LM progress is so fast, that static evaluation tasks are at risk of quickly becoming obsolete. Ideally, we would like our tasks to co-evolve in difficulty and sophistication with the models they will evaluate.

We believe the above desiderata represent necessary conditions to ensure we remain able to evaluate language models during this new era. In our work, we make the argument that structured negotiation games represent a particularly suitable construct. Specifically, they:
1. Occur as a realistic downstream task ubiquitous in modern society
2. Are simple to implement and extend to arbitrary levels of difficulty
3. Allow us to jointly evaluate performance and metrics relevant to alignment
4. Enable cross-model interaction measurements in a competitive and collaborative setting
5. Parameterizing tasks using LMs side-steps data leakage and co-evolves task complexity

We present a scalable, core implementation of our method and empirically tested its efficacy on LMs from all major public providers, including the current state-of-the-art open-source LLaMA 2 models. We took great care in controlling for bias and pointing out additional considerations and potential limitations. We further showed that even on ‘simple’ negotiation games, current LM-agents struggle – this is a good thing! Our benchmarking approach is meant to be forward-looking and present an ongoing challenge.

Finally, we believe it is crucial to encourage research in this important direction and lower the boundaries for researchers outside of machine learning to contribute. To that end, we open source our code implementation, which is designed to require minimal to no additional coding efforts from its end-users. We also plan to release the thousands of negotiation transcripts generated during the development of this work as a publicly accessible dataset. This large collection of cross-model interactions will be the first of its kind.

---

### Meta-Review · Area_Chair_vdRe · 2023-12-09

**Metareview:**

a) claims: The paper proposes to evaluate the agency of LLM models by having them negotiate against each other.  The most popular commercial models are evaluated, as well as the leading open-source models.

b) strengths: The reviewers all agree that the basic idea of using negotiation ability as a proxy for general agency is promising.  Some noted that evaluating widespread general agency claims for LLMs is a timely and relevant question.

c) weaknesses: Both negative reviews commented on clarity issues with the paper; the central concept of agency was not defined in a way that connected well with the main exercise of the paper.  There were also concerns about the experimental procedure: the experiments are less reproducible than would be ideal due to the closed nature of the commercial models being evaluated, and the need for manual intervention in the procedure.

**Justification For Why Not Higher Score:**

I have some concerns about the clarity of central definitions that were raised in review.

**Justification For Why Not Lower Score:**

The ideas are important and I think they will lead to important discussions; all the reviewers agree that this is a promising and interesting approach to an important question.

---

### Decision · Program_Chairs · 2024-01-16

Accept (poster)